# Role of Dipeptidyl Peptidase-4 (DPP4) on COVID-19 Physiopathology

**DOI:** 10.3390/biomedicines10082026

**Published:** 2022-08-19

**Authors:** Alba Sebastián-Martín, Belén G. Sánchez, José M. Mora-Rodríguez, Alicia Bort, Inés Díaz-Laviada

**Affiliations:** 1University of Alcalá, School of Medicine and Health Sciences, Department of Systems Biology, Biochemistry and Molecular Biology Unit, 28871 Madrid, Spain; 2Department of Comparative Medicine, School of Medicine, Yale University, New Haven, CT 06519, USA

**Keywords:** DPP4, CD26, type 2 diabetes, inflammation, SARS-CoV-2, COVID-19, gliptins

## Abstract

DPP4/CD26 is a single-pass transmembrane protein with multiple functions on glycemic control, cell migration and proliferation, and the immune system, among others. It has recently acquired an especial relevance due to the possibility to act as a receptor or co-receptor for SARS-CoV-2, as it has been already demonstrated for other coronaviruses. In this review, we analyze the evidence for the role of DPP4 on COVID-19 risk and clinical outcome, and its contribution to COVID-19 physiopathology. Due to the pathogenetic links between COVID-19 and diabetes mellitus and the hyperinflammatory response, with the hallmark cytokine storm developed very often during the disease, we dive deep into the functions of DPP4 on carbohydrate metabolism and immune system regulation. We show that the broad spectrum of functions regulated by DPP4 is performed both as a protease enzyme, as well as an interacting partner of other molecules on the cell surface. In addition, we provide an update of the DPP4 inhibitors approved by the EMA and/or the FDA, together with the newfangled approval of generic drugs (in 2021 and 2022). This review will also cover the effects of DPP4 inhibitors (i.e., gliptins) on the progression of SARS-CoV-2 infection, showing the role of DPP4 in this disturbing disease.

## 1. Introduction

The dipeptidyl peptidase 4 (DPP4, EC 3.4.14.5) was described for the first time in 1966 [1]. It is also known as T-cell activation antigen CD26 [2], or adenosine deaminase binding protein (ADBP) [3]. Its diverse terminology reflects the multifunctional or moonlighting character of the protein. The functions of such kind of proteins depend on their intracellular or extracellular localization, cell type, monomeric or oligomeric state, and concentration of substrates and ligands [4]. In the case of DPP4, its actions do not only rely on its catalytic activity as a peptidase, but also on its own structure, given that DPP4 can bind to several proteins, like adenosine deaminase (ADA), fibronectin, collagen, chemokine receptor CXCR4, tyrosine phosphatase CD45, and even some viral proteins such as the Human Immunodeficiency Virus (HIV) gp120 envelope protein. Thus, it regulates multiple cellular processes, playing a role in adhesion to the extracellular matrix, proliferation, and in T-cell maturation and activity.

Recently, DPP4 has acquired certain relevance in the scenario of the severe acute respiratory syndrome coronavirus 2 (SARS-CoV-2) infection, due to its potential role as a cellular entry receptor or co-receptor for the virus. Although this hypothesis needs to be further elucidated, data from clinical studies indicate that DPP4 participates in the coronavirus disease 2019 (COVID-19) physiopathology and therefore is a therapeutic target for this disease. The most prevalent comorbidities in SARS-CoV-2-infected patients were hypertension and diabetes, followed by cardiovascular diseases and respiratory system disease [5]. Interestingly, DPP4 has a striking role in these disorders, especially on type 2 diabetes mellitus (T2DM). 

This review aims to update the field of DPP4 and COVID-19, trying to unravel the putative mechanisms by which the protease plays a role in the course of the infection by SARS-CoV-2. In this sense, the structure and the physiological activities of DPP4 will be outlined, focusing on endogenous glycemic control and the immune system, due to their implications on COVID-19 progression. Afterwards, we will explore the effect of DPP4 during the SARS-CoV-2 infection. For that purpose, it is interesting to firstly analyze the evidence around the role of DPP4 on viral entry, not only in SARS-CoV-2, but also in previous coronaviruses. Secondly, the striking impact of T2DM and inflammation will be exposed in the context of COVID-19. To do so, both aspects will be introduced, taking into account their relationship with DPP4 to connect them with the prognosis on positive patients. Finally, we offer an update on the DPP4 inhibitors approved by the European Medicine Agency and the U.S. Food and Drug Administration, ending with the effect of the said inhibitors on COVID-19 patients.

## 2. Materials and Methods

A systematic review was conducted using the following search engines: MEDLINE/PubMed, SCOPUS, Web of Science, and Google Scholar. Medical Subject Headings (MeSH) terms such as “COVID-19”, “SARS-CoV-2”, “type 2 diabetes mellitus and COVID-19”, “hyperglycemia and COVID-19”, “dipeptidyl peptidase 4 structure”, “Dipeptidyl peptidase 4 inhibitors”, and “DPP4 inhibitors and COVID-19” were used. 

## 3. Results 

### 3.1. Structure of DPP4 

DPP4 is a type II transmembrane glycoprotein [6] of 766 amino acids [7,8] and 110 kDa of molecular weight [9]. It is a serine exopeptidase belonging to the clan SC, family S9, and specifically to the S9B protein subfamily, based on the hierarchical and structural criteria of the MEROPS database [10]. Peptidases of the SC clan present an atypical sequential order of the catalytic residues Ser-Arg-His (in the case of DPP4, at positions 630, 708, and 740), instead of His-Arg-Ser as in the classical serine proteases [7,11,12]. The S9B subfamily is characterized by the motif Gly-Trp-Ser-Tyr-Gly-Gly-Tyr around the Ser630 located in the active site [13]. Other members belonging to this subfamily are fibroblast activation protein (FAP), resident cytoplasmic proteins (DPP8 and DPP9), and non-enzymatic members such as DPP6 and DPP10 [6,7,8,9]. They are also known as DPP4 activity and/or structure homologue (DASH) proteins.

The primary structure shows an N-terminal cytoplasmic domain of six amino acids (aa), a transmembrane segment of 22 aa, and a larger extracellular part of 738 aa [14,15,16,17,18]. In turn, the extracellular part comprises a short flexible stalk (aa 29–39), a glycosylation-rich region (aa 101–350), a cysteine-rich region (aa 55–100, 351–497), and a catalytic domain (aa 506–766) [19] (Figure 1A). Analysis of the crystal structure of DPP4 distinguishes two domains in the extracellular part: a β-propeller and a catalytic α/β hydrolase domain [16,20]. The β-propeller is open, has eight blades [21] and encompasses two subdomains: the glycosylation-rich region (blades II–V) and the cysteine-rich region (blades VI–VIII) [8,15,16,22]. The regions constituting the β-propeller could serve to interact with other proteins and be involved in non-enzymatic functions of DPP4 [23]. For example, ADA binds to the glycosylation-rich region of DPP4, and matrix proteins such as collagen to the cysteine-rich region [17,19,20,22,24]. On the other hand, the α/β hydrolase domain harbors the catalytic triad (Ser630/Asp708/His740) required for DPP4 activity [20,25,26]. Two openings can be distinguished in the DPP4 monomer through which the active site could be accessed, (i) a lateral one formed by the β-propeller, and the α/β hydrolase domain; and (ii) a smaller one constituted by the β-propeller domain [19,21]. It is suggested that the entry of the substrate into the active center occurs through the side entry since the peptide in this way would be correctly oriented for cleavage and because the exit of the resulting peptide occurs through the funnel localized in the middle of the β-propeller domain [27,28] (Figure 1B). 

During the translation, the typical signal peptide that allows DPP4 to be driven to the endoplasmic reticulum is also needed to initiate the translocation across the membrane and further serves as membrane anchor [9]. Then, DPP4 can undergo several modifications, like N-glycosylation [9], oxidation [29], sialylation, and phosphorylation [15]. N-glycosylation has been linked to folding and stability, while sialylation at the N-terminal end has been related to the trafficking of DPP4 to the cell apical membrane [15]. Once in the membrane, DPP4 usually forms dimers, which is considered a prerequisite for carrying out its enzymatic activity [6,9]. Normally, two DPP4 proteins are located close together and form a U-shaped homodimer [20,30]. The arms of the U are formed by β-propeller domains and are located distal to the plasma membrane. The curvature of the U, which is located proximal to the membrane, is constituted by the α/β-hydrolase domains that harbor the catalytic triad [20] (Figure 1B).

DPP4 protein can be cleaved from the membrane by metalloproteases (MMPs), yielding to a process called shedding [9,22,23]. (Figure 1B) As a consequence, the soluble and catalytically active form of DPP4 is released and migrates in fluids such as serum, saliva, bile, cerebrospinal fluid, and semen [22,23]. Soluble DPP4 (sDPP4) usually circulates as a dimer although it can also assemble and form larger complexes [24,31]. This soluble form of DPP4 is given an immunoregulatory role [32] and is thought to activate cell signaling pathways, although the mechanisms involved remain unknown [8]. Serum levels of sDPP4 have been linked to a multitude of diseases [22].

The expression of DPP4 is regulated at the molecular level. It has been described that the promoter region of the DPP4-enconding gene possesses a G and C-rich region. This serves as a binding site for transcription factors such as NF-kB, EGFR, AP-1, among others, which participate in the regulation of protein expression [9,19]. The participation of several cytokines such as IL-12 [19], and other transcription factors such as HIF-1α [9], has also been described in this regulation. Regarding its localization, the DPP4 protein is usually ubiquitously distributed throughout the organism [7,24]. It has been localized in the epithelium and endothelium of a large number of organs [15] such as kidney, lung, liver, intestine, brain, heart, prostate, pancreas, and skeletal muscle [7,9,19,24,32,33]. It is also expressed in immune cells [15], as described below. 

### 3.2. Physiological Role of DPP4

This enzyme catalyzes the digestion of multiple chemokines, neuropeptides, and regulatory peptides, preferentially when containing a proline residue at the penultimate position of the amino-terminal region, releasing a dipeptide. However, DPP4 also cleaves peptides bearing alternative residues at position 2, such as hydroxyproline, dehydroproline > alanine, glycine, threonine, valine, or leucine [8], although only oligopeptides in the *trans* conformation are able to bind to the active site [13]. The resulting inactivated or new bioactive peptides answer for the diverse biological processes that DPP4 regulates. Many approaches have tried to determine the DPP4 degradome. Those methods include pharmacological in vitro assays, where putative DPP4 substrates are incubated with purified sDPP4, extracts of cells expressing endogenous or transfected DPP4, or plasma containing DPP4. In addition, there are more challenging physiological in vivo experiments, in which native substrates are studied in animals or humans treated with DPP4 inhibitors (reviewed in [8]). Peptides that showed a difference between the intact and cleaved fraction, in the presence or absence of DPP4, are considered substrates of the enzyme. 

Several of the best-known substrates of DPP4 include incretins, substance P, neuropeptide Y, stromal cell-derived factor 1α/β (SDF-1α/β), granulocyte-macrophage colony-stimulating factor (GM-CSF), CXCL10, brain natriuretic peptide (BNP), and pituitary adenylate cyclase-activating polypeptide (PACAP) [34]. However, the functions of DPP4 do not only rely on its hydrolyzing activity, but also on its own structure, by interacting with multiple factors (Figure 2). Thus, overall, DPP4 performs multiple activities in metabolism, cardiovascular system, immunology, endocrinology, fibrosis, and cancer [35]. It is related to cellular processes like glycemic control, cell migration and proliferation, or the immune system and associated inflammatory processes. 

#### 3.2.1. Physiological Role of DPP4 in Glycemic Control

DPP4 is involved in the endogenous control of glycemia, being physiologically or pharmacology related to the degradation of glucagon, glucagon-like peptide-1 (GLP-1) and -2 (GLP-2), gastric inhibitory peptide or glucose-dependent insulinotropic polypeptide (GIP), and gastrin-releasing peptide (GRP) [8]. Physiologically, after oral meal intake, several gastrointestinal hormones called incretins are secreted into the bloodstream to allow the improvement of peripheral glucose tolerance by stimulating postprandial insulin secretion in the pancreas. More than 90% of incretin activity is performed by GLP-1 and GIP [36]. GIP is produced predominantly in duodenal K cells in the proximal small intestine, whereas GLP-1 is secreted from distal intestinal L cells, rather before the digested nutrients cross the small bowel to contact directly with these enteroendocrine cells, so neural and endocrine factors are expected to promote GLP-1 secretion [37]. Both GLP-1 and GIP promote pancreatic β cell proliferation and inhibit apoptosis, contributing to the expansion of these cells, and to the insulin secretion they produce [38]. Interestingly, both incretins contain an alanine residue at position 2 in the N-terminal end, and they have been demonstrated as physiological endogenous substrates of DPP4 [8]. Plasma levels of intact active GLP-1 and GIP are increased in *Dpp4*^−/−^ mice [39], as well as in animals and humans treated with DPP4 inhibitors [39,40,41,42]. Fisher344/DuCrj rats with reduced DPP4 activity also showed increased levels of intact GLP-1 [43]. 

Inactivation of these incretins via DPP4 digestion occurs quickly, given that their half-lives have been estimated around 1–2 min for GLP-1 and 7 min for GIP [44,45], along with the kidney clearance. GIP is a 42-amino acid hormone, being active in its full-length GIP_1–42_ and inactive when truncated as GIP_3–42_ by DPP4. On the other hand, GLP-1 proceeded from proglucagon, which in intestinal L cells, is cleaved by the prohormone convertase 1 (PC1), also known as PC1/3 or PCSK1. The resulting GLP-1 forms include full-length GLP-1_1–37_ (or GLP-1_1–36amide_) which has lower insulinotropic efficacies, as well as the truncated GLP-1_7–37_ and GLP-1_7–36amide_, which are potent stimulating insulin secretion [46,47,48,49]. In humans, ~80% of circulating active GLP-1 is 7–36 amide, and ~20% the 7–37 form [50]. In that case, DPP4-mediated digestion takes place over the already PC1-truncated forms of GLP-1, rendering GLP-1_9–36amide_ and GLP-1_9–37_, ligands with low affinity for the GLP-1 receptor [51,52]. In the intestine, DPP4 is highly expressed in the brush border epithelium as well as in the endothelial cells, suggesting that GLP-1_7–37_ and GLP-1_7–36amide_ are digested in the capillaries of the distal gut with only around 10–15% of active GLP-1 reaching the bloodstream [53,54]. The GLP-1 metabolites yielded by DPP4 cleavage have no major roles in glucose metabolism, glucose clearance, or insulin secretion in healthy humans. In spite of that, in one experiment in obese humans, it has been reported that GLP-1_9–36amide_ is a weak insulin segretagogue, and its administration improves glucose handling without affecting insulin secretion (reviewed in [54]). 

Thus, DPP4 limits the action of incretins that, when are intact, arrive through the circulation to the pancreas. Briefly, GLP-1 contacts in the pancreas with its receptor (GLP-1R), a G-protein-coupled receptor predominantly localized in β cells. This connection activates the adenylate cyclase (AC), stimulating cyclic AMP formation and subsequently the protein kinase A (PKA), along with signaling via exchange proteins activated by cAMP (i.e., Epac), such as cAMP-guanine nucleotide exchange factor (GEF)-II. Consequently, voltage-gated K^+^ (K_v_) and K_ATP_ channels are inhibited, while L-type voltage-dependent Ca^2+^ channels are opened, leading to the increase of intracellular calcium that promotes exocytosis of the insulin granules and its acute secretion into the circulation [54]. Furthermore, the GIP receptor is also found in β cells and its activation is also coupled to AC, cAMP, and Ca^2+^ influx, which produce the release of insulin [55]. GIP signaling also proceeds through the extracellular signal-regulated kinases 1 and 2 (ERK 1/2), phosphatidylinositol 3-kinase (PI3K), and protein kinase B (Akt), among others. 

#### 3.2.2. Physiological Role of DPP4 in the Immune System 

Besides the effects on glucose homeostasis by its enzymatic activity, DPP4 has pleotropic immune regulatory actions mostly mediated by protein–protein interactions. In fact, DPP4 is expressed in many types of immune cells including T cells, B cells, natural killer cells (NKs), dendritic cells (DCs), and macrophages [56]. In this line, DPP4 has many recognized partners, with implications in the immune system and T cell function, such as adenosine deaminase (ADA), caveolin-1, the protein tyrosine phosphatase CD45, CXCR4, collagen (especially 1 and 3), fibronectin, glypican 3, caspase-recruitment domain containing 1 (CARMA-1), or the mannose 6-phosphate/insulin-like growth factor II receptor (M6P/IgFr2) ([22] and references herein). 

One of the most relevant partners of DPP4 is ADA, given that its presence assures the functionality of T, B, and NK cells, as well as an adequate cellular and humoral immunity. ADA deficiency causes early-onset severe combined immunodeficiency, increasing the susceptibility to suffer from infections [57,58]. The presence of DPP4 in the membrane of immune cells permits the accumulation of ADA on the cell surface, by acting as an ADA-anchoring factor. ADA also has implications in inflammation, as we will described below. On the other hand, the interaction of DPP4 with CD45 occurs by its intracellular domain, causing the recruitment of both enzymes on lipid rafts and facilitating the co-localization of CD45 and T cell receptor (TCR) signaling molecules. These facts inhibit CD45 dimerization and activity, thereby enhancing protein tyrosine phosphorylation of various signaling pathways with key roles in T cell activation [59]. Activation of peripheral blood T cells results in the subsequent mannose-6-phosphorylation of DPP4. Then, DPP4 can bind to the M6P/IgFr2 receptor that induces its internalization, playing an important role in DPP4-mediated T cell co-stimulation [60]. Furthermore, DPP4 interacts with caveolin-1 on antigen-presenting cells, enhancing CD86 expression, T-cell proliferation, and NF-κB activation. Overall, DPP4 is considered a marker protein for T cell function [22]. 

### 3.3. DPP4 in COVID-19

#### 3.3.1. Interaction between Coronaviruses and DPP4

Human-infective coronaviruses utilize host surface cellular receptors to bind and infect the mammalian cell. In this sense, cellular entry depends on the binding of the spike glycoprotein (S) to a specific cell membrane protease that acts as a receptor, along with the subsequent cleavage of S glycoprotein at two sites by distinct proteases, firstly by Furin and then by the type II transmembrane cellular protease serine 2 (TMPRSS2). The S spike protein is composed of two domains: the S1, responsible for host cell receptor recognition; and S2, which mediates virus entry into the cell. The S1 domain contains the receptor-binding domain (RBD) while the S2 subunit is the one that inserts into the cell membrane containing a single-pass transmembrane anchor, and a short intracellular tail. Once RBD interacts with the receptor, Furin cuts the S protein at the S1/S2 interaction, dividing S1 and S2, and then, TMPRSS2 cleaves the S2 domain. Cleavage releases the constriction that S1 exerts on S2 and relaxes the structure of S2, allowing the formation of a fusion protein that inserts into the cell membrane inducing endocytosis of the entire viral particle. Viral entry is also dependent on the activity of cathepsin B/L which can be a substitute for TMPRSS. Then, virus recognition occurs when the S protein interacts with the membrane protease, which facilitates membrane fusion at the plasma membrane [61,62].

Comparing the Spike S protein of the coronaviruses SARS-CoV-2, SARS-CoVs, and MERS-CoV, the longest S protein is that of MERS-CoV with a further 80aa than the SARS-CoV-2 S protein, which in turn is 18 aa longer than that of SARS-CoV [63]. Most of the variation relies on the S1 subdomain, preferentially in the N-terminal portion. SARS-CoV-2 S1 contains a single 4-aa insertion and six deletions compared to the SARS-CoV-2 S1 subdomain, whereas MERS-CoV harbors 19 insertions and 13 short deletions [63]. A multiple sequence alignment of the spike protein of each virus demonstrated that MERS-CoV showed 30.63% homology to SARS-CoV and SARS-CoV-2 displayed 76.19% homology to SARS-CoV [64]. Interestingly, the RBD SARS-CoV-2, SARS-CoVs, and MERS-CoV display a similar architecture, consisting of a β-sheet core with a twisted five-stranded antiparallel β sheet and an inserted loop that directly interact with the receptor [65]. However, SARS-CoV-2-S RBD is more structurally similar to MERS-CoV RBD than SARS-CoV RBD to MERS-CoV RBD [66]. Both SARS-CoV-2 and MERS-CoV S proteins have a polybasic cleavage site at S1/S2, which is absent in the SARS-CoV S protein, that remains uncleaved during assembly and exocytosis. The S2 domain is 14aa longer in MERS-CoV-2 than in either Sars-CoV-2 or SARS-CoV, which are equally long and show ~91% identity [67].

Remarkably, all host surface receptors identified to date for the human-infecting coronaviruses are exopeptidases (i.e., angiotensin-converting enzyme 2 (ACE2), dipeptidyl peptidase 4 (DPP4), and aminopeptidase N (APN) [68]), although the ectopeptidase function does not appear to be required for viral entry [69]. The main reported functional receptor for SARS-CoV-2 is the protein ACE2, the same receptor for SARS–CoV. In addition, other peptidases have also been proposed as receptors for coronaviruses. Notably, DPP4 was identified as the main functional receptor for Middle East respiratory syndrome coronavirus (MERS-CoV) [7] [initially named human coronavirus-Erasmus Medical Center (hCoV-EMC) [8]]. DPP4 was specifically co-purified with the Spike protein’s RBD of MERS-CoV in lysates of susceptible Huh-7 cells. Interaction between MERS-CoV S protein and DPP4 was essential for viral infection and correlated with susceptibility to MERS-CoV infection, as well as with viral genome detection in the culture medium of infected cells [70]. Conversely, computational analysis showed that according to the free energy values, there were no potential interactions of SARS-CoV S protein with DPP4 (free energy value > 0 kcal/mol) [66].

It is worth noting that DPP4 residues involved in virus–protease interaction are highly conserved between humans and bats. However, an experimental demonstration of the interaction between SARS-CoV-2 and DPP4 has not yet been reported, and in vitro studies could not demonstrate that SARS-CoV-2 RBD binds to DPP4-expressing cells [71]. Nevertheless, several findings suggest that SARS-CoV-2 coronavirus does not exclusively use ACE2 and might rely on other receptors for cellular entry [72]. For instance, the expression of ACE2 in the lung is relatively low whereas the lung is one of the main tissues suffering important damage in COVID-19 [73]. Likewise, ACE2 expression decreases with age although COVID-19 risk and severity increase with age [74]. In line with this, recent evidence suggests that SARS-CoV-2 may be using DPP4 as a co-receptor when entering the cells [67]. In fact, DPP4 is expressed in the primary tissues involved in viral infection susceptibility, since it is highly expressed in the alveolar type 2 (AT2) cells of the distal lung, as well as on the surface of the epithelium, vascular endothelium, and fibroblasts of human bronchi, where it plays a role in different lung diseases [75]. A recent study showed that the protease TMPRSS2 and DPP4 were co-localized in limbal and corneal epithelial superficial cells [76], a proposed SARS-CoV-2 entryway. In placenta cells, minimal expression of both ACE2 and TMPRSS2 and a high expression of DPP4 was found, despite the placenta being susceptible to SARS-CoV-2 infection [77]. Likewise, while ACE2 levels are very low in cortical cells, a high DPP4 expression was found in cortical astrocytes infected with SARS-CoV-2, and interestingly, DPP4 inhibition reduced viral infection [78]. 

Despite lacking definitive demonstrations that DPP4 is a receptor for SARS-CoV-2, bioinformatics approaches combining human–virus protein interaction prediction and protein docking based on crystal structures have revealed the high affinity between human DPP4 and the spike S RBS of SARS-CoV-2 [66]. In fact, DPP4 ranked in the top five putative human receptors for SARS-CoV-2 in the predictive analysis and had the highest score for protein–protein interaction. The crucial residues of DPP-4, essential for binding to the SARS-CoV-2 spike protein, were identical to those described for binding to protein S of MERS-CoV [66]. Molecular docking computational analysis of the interaction between the SARS-CoV-2 S glycoprotein and DPP4 showed a large interface between the proteins, suggesting a tight interaction between them [67]. Moreover, this study identified 14 critical binding residues for interaction of DPP4 with SARS-CoV-2 S glycoprotein [67] (five more than with MERS-CoV S protein) [79], suggesting a strong binding between these two proteins. On the other hand, it has been described that a Neanderthal variant of the DPP4 promoter doubles the risk of being hospitalized for COVID-19, with DPP4 levels correlating with the severity of COVID-19 [80]. 

Crystal structures of the RBD of the spike protein complexed with DPP4 demonstrated that the interaction does not occur at the protease catalytic domain but at the DPP4 β-propeller domain, suggesting that the virus–receptor interaction is independent of the peptidase activity [81]. However, the interaction was very similar to the binding of DPP4 with one of its main partners (i.e., ADA); and all those DPP4 residues identified in the virus–protease engagement were also involved in ADA binding [81,82]. Accordingly, ADA competed with MERS-CoV for binding to DPP4, acting as a virus–DPP4 attachment inhibitor and preventing virus infection [83]. This is in good agreement with the physiological role of DPP4 in immune cells, recognizing ADA protein (see below). This observation guides through the suggestion that DPP4 could be a co-receptor or auxiliary protein to facilitate the virus entry, as a kind of ACE2 partner. This means that the coronavirus SARS-CoV-2 might use multiple receptors to enter host cells [84]. Such co-receptors might help virus internalization, increasing intracellular viral load as well as driving and exacerbating immunopathological hyperinflammation. In fact, single cell transcriptomic analysis of 13 human tissues found similar mRNA expression profiles of ACE2 and DPP4 [85]. Interestingly, a strong positive correlation between DPP4 localization and the site of lung inflammation has been observed. Computational research for human DPP4′s nearest neighbor proteins showed that DPP4 interacts with ACE2, implying a cross-talk between both proteins [86]. However, more studies are needed to know how ACE2 and DPP4 interact and how DPP4 may help the entry of SARS-CoV-2 into host cells.

In order to highlight the importance of DPP4 in COVID-19, two monoclonal antibodies targeting this protease have been designed to treat COVID-19 patients. Begelomab, a monoclonal anti-DPP4 antibody was efficacious treating hematological conditions that mimic the hyper-inflammation caused by the coronavirus SARS-CoV-2 [87]. A new antibody was designed to interact with both ACE2 and DPP4 but its use in patients has not been assessed yet [88]. Although no clinical trials have been conducted to date to evaluate their efficacy, anti-DPP4 antibodies might be considered an interesting approach, worth being tested for dealing with COVID-19. 

#### 3.3.2. DPP4 and Diabetes in COVID-19 Patients

DPP4 and Diabetes Mellitus Type 2

Type 2 diabetes mellitus (T2DM) has been established as the most important risk factor for SARS-CoV-2 infection [89,90]. T2DM is a heterogeneous disease characterized by elevated levels of blood glucose as a result of peripheral insulin resistance (IR), which is the impaired ability of target tissues to sense or respond to insulin stimulation, together with varying degrees of deficient insulin secretion by pancreatic islet β cells (β cell dysfunction) [91]. Inadequate compensatory insulin secretory responses are also associated with this process [92]. T2DM has been commonly known as a non-insulin-dependent condition, in contrast to diabetes type 1 in which an absolute insulin deficiency is produced, associated with autoimmune destruction of the pancreatic β cells [93]. It has been shown that GLP-1, unlike GIP, potently stimulates insulin secretion and reduces blood glucose in human subjects with T2DM [37,38]. 

In this context, DPP4 inhibitors have been explored as candidates for preventing the inactivation of GLP-1 and GIP. They were studied in both preclinical and clinical studies [94,95,96], and later approved as oral drugs for the treatment of T2DM [97]. DPP4 inhibitors are also known as gliptins and sometimes referred to as incretin enhancers. They are recommended as second line therapy after metformin for managing T2DM by the consensus of the American Diabetes Association and the European Association for the Study of diabetes [98,99], and mainly used as add-on to metformin. For a review of the effects of DPP4 inhibitors, also in combination with other antidiabetic drugs, see [100]. 

As pointed above, T2DM is characterized by hyperglycemia and insulin resistance. The high plasma glucose promotes the synthesis of advanced glycation end products (AGEs) which induce impairment of the immune system through binding to their receptors (i.e., RAGE). This alteration includes neutrophil dysfunction, leukocyte recruitment inhibition, suppression of cytokine production, defects in phagocytosis, and disability of immune cells to control invading pathogens [101]. Consequently, diabetic patients are more susceptible to infection than non-diabetic patients. In addition, in T2DM, there is a change in the immune system cells, which shift from an anti-inflammatory to a predominantly chronic pro-inflammatory stage. Moreover, in blood, AGEs behave as cross-linkers, increasing cell adhesion and vascular stiffness. In the context of hyperlipidemia and hypertension usually associated with T2DM, these mechanisms often lead to cardiovascular disease [102]. 

Moreover, sDPP4 might contribute to endothelial dysfunction as it has been shown that DPP4 impaired the endothelium-dependent relaxation elicited by acetylcholine in a concentration-dependent manner [103]. Interestingly, AGE-induced generation of reactive oxygen species stimulates the release of DPP4 from endothelial cells, which could in turn act on endothelial cells directly via the interaction with M6P/IgFr2, further potentiating the deleterious effects of AGEs [104].

Impact of DPP4 and Diabetes on SARS-CoV-2-Positive Patients

As the most important risk factor for SARS-CoV-2 infection, T2DM also impacts diabetic patients with higher mortality rates [89,90]. Coronavirus infection notably increased mortality in diabetic patients, as shown in a retrospective study including 6014 subjects with diabetes in which those positive for COVID-19 infection were 3.46 times more likely to die than those who tested negative [90]. Besides, diabetes mellitus is one of the main comorbidities of COVID-19 [105,106,107]. It is interesting that, conversely, a new-onset of diabetes with metabolic dysregulation and impaired glucose homeostasis, as well as severe metabolic complications, has been described as a consequence of SARS-CoV-2 infection [108]. 

A recent study to evaluate the effect of hyperglycemia and hypercoagulability on COVID-19 prognosis has shown that elevated hyperglycemia and D-dimer had a synergistic effect on COVID-19 prognosis, and this risk was independent of diabetes history [109]. In the case of COVID-19, in which physiopathology is characterized by hyperinflammatory response with cytokine overproduction and cardiovascular disorder, it is not yet clear whether the diabetes condition increases the risk of infection or its severity [110], magnifying the pathogenicity of SARS-CoV-2 since both COVID-19 and diabetes share some pathological mechanisms. 

High glucose levels might enhance the expression of SARS-CoV-2 receptors in the surface of cells, increasing the risk of infection. A retrospective study examining COVID-19 heart autopsies, revealed that total ACE2, glycosylated ACE2, and TMPRSS2 protein expressions were higher in cardiomyocytes from autopsied and explanted hearts of diabetic than non-diabetic samples [111]. On the other hand, high glucose increases the synthesis and secretion of DPP4 in liver [112]; and plasma DPP4 levels and activity were increased in T2DM patients compared to controls [113]. Increased levels of sDPP4 have also been detected in metabolic syndrome, where sDPP4 positively correlated with various parameters such as body mass index, adipocyte surface, and leptin and insulin levels. Moreover, DPP4 expression in visceral adipose tissue, as well as sDPP4, is increased in obese patients, contributing to T2DM physiopathology [114]. The high expression of DPP4 in the visceral adipose tissue from where it can be released and contribute to sDPP4 has made DPP4 to be considered as a novel adipokine [115]. 

Independent of the potential role of DPP4 as SARS-CoV-2 receptor, DPP4 might have a role in COVID-19 incidence and physiopathology by regulating glucose homeostasis. A multicenter retrospective study determined that blood glucose, gender, prothrombin time, and total cholesterol could be considered risk factors for COVID-19 [116]. Another investigation with more than 2000 cases concluded that elevated glucose was an independent risk factor for progression to critical cases or death in COVID-19 inpatients [117]. In addition, it has been observed that COVID-19 patients with elevated levels of glycated hemoglobin HbA1c had more severe inflammation and higher mortality than patients with normal levels. Even in patients with only elevated HbA1c level but no diabetes, the levels of inflammation markers were also significantly increased, pointing to a role of persistent high glycemia in the physiopathology of COVID-19 [118]. Conversely, a study about the association between glucose control of COVID-19 patients with T2DM revealed that well-controlled blood glucose levels in the first 7 days could improve the prognosis of COVID-19 inpatients with diabetes [119]. 

Mechanistically, it could be possible that a hyperglycemic state might favor virus infection as it has been shown that elevated glucose levels enhanced SARS-CoV-2 replication and cytokine expression in monocytes through stabilization of HIF-1α [120]. Authors conclude that the high glucose availability might prime glycolysis needed for virus replication and for monocyte immune response. Moreover, glucose levels may affect the glycosylation patterns of the virus spike protein modifying its conformation, and reducing its binding to ACE2, as it has been confirmed through biochemical experiments [121]. Altogether, these findings indicate that high glucose levels might favor virus replication and immune dysregulation and that DPP4 could participate in the worst clinical findings shown in COVID-19 diabetic patients, through regulation of both phenomena. 

One question that arises is whether hyperglycemia can impact the glycosylation of DPP4, modifying its activity and dimerization. The primary structure of DPP4 contains nine N-glycosylation sites [122]. The correct glycosylation of DPP4 is a requisite for enzyme activity, along with proper protein folding and accurate trafficking [122], whereas aberrant glycosylation has been associated with pathological processes. For instance, glycosylation at the Asn520 site has been detected in Kashin–Beck disease, a chronic deformative osteoarthropathy and might contribute to cartilage destruction [123]. Mutation of the sixth N-glycosylation site of rat DPP4 abolished the enzymatic activity, eliminated cell-surface expression, and prevented the dimerization of the DPP4 protein [124]. Moreover, four glycosides linked to the conserved Asn229 participate in the interaction of DPP4 with ADA [20]. Although there is no experimental evidence suggesting that hyperglycemia-induced aberrant glycosylation of DPP4 may play a role in COVID-19 physiopathology, several pieces of information shed light on this hypothesis. Glycosylation of mouse DPP4 at Thr330, a non-conserved glycosylation site, is a substantial barrier to MERS-CoV infection, but DPP4 may act as a virus receptor when glycosylation is absent [125]. 

#### 3.3.3. DPP4 and the Immune Response in COVID-19 Patients

DPP4 during inflammation

On the cell surface, DPP4 interacts with several proteins including adenosine deaminase (ADA). ADA catalyzes the irreversible deamination of adenosine and 2′-deoxyadenosine to inosine and 2′-deoxyinosine, decreasing adenosine levels and blocking biological effects of adenosine. When present, adenosine binds to A1, A2A, A2B, and A3 receptors belonging to the G-protein-coupled receptors superfamily, which are expressed on the surface of most immune cells and modulate many aspects of the immune responses, essentially immunosuppressive and anti-inflammatory [126]. Moreover, it has been recently described that adenosine exerts a key role in the inflammatory resolution governing processes to promote the clearance of inflammatory cells and a return to local tissue homeostasis such as the interruption of leukocyte infiltration, the counter-regulation of pro-inflammatory mediators, the uptake of apoptotic neutrophils and cellular debris, and the repolarization of the immune cell phenotype [126]. By binding to ADA through its extracellular domain, DPP4 recruits the enzyme on the surface of lymphocytes, reducing adenosine levels and preventing the effect of adenosine in situ.

In addition, and independent of its enzymatic activity, ADA accomplishes with DPP4 by the extracellular side in a complex formed by two ADA molecules and a DPP4 dimer. ADA association with DPP4 on the T cell surface allows the interaction of ADA with adenosine receptors, forming a ternary complex that is thought to be important, as a costimulatory signal to promote proliferation of lymphocytes and cytokine production, leading to a marked increase (3- to 34-fold) in the production of the pro-inflammatory cytokines IFN-γ, TNF-α, and IL-6 [127]. Altogether, these observations indicate that DPP4 is an important regulator of the immune system and therefore can contribute to the comorbidities of COVID-19.

Impact of DPP4 and inflammation at the COVID-19 clinics

One of the characteristics of COVID-19 is an amplified and aberrant immune response to SARS-CoV-2 infection, resulting in a cytokine storm, potentially triggering acute lung injury, and leading to the acute respiratory distress syndrome which gave the name’s disease. The invasive inflammatory response releases large amounts of pro-inflammatory cytokines, causing uncontrolled systemic inflammation, which contribute to physiopathology and eventually lead to fibrosis, causing tissue damage and organ failure. In addition, coronary plaque destabilization, and hypoxia induce damage of cardiomyocytes [128]. Notably, cardiovascular damage and coagulopathies contribute to the complications of COVID-19. 

It is remarkable that serum DPP4 levels and activity were significantly lower in COVID-19 patients at hospital admission compared to healthy controls [129]. Moreover, a significant decrease in serum DPP4 activity was found in COVID-19 inpatients, which was associated with severe COVID-19 disease and mortality [130]. In addition, a significant reduction in serum DPP4 levels was seen in relation to T2DM, age, and age-related dementia [131]. Authors propose that high serum DPP4 levels could protect from viral infection by competitively inhibiting the virus binding to cellular DPP4, whereas low serum DPP4 levels could increase the risk of infection [131].

It is worth noting that it has been demonstrated that glucocorticoids, like dexamethasone, by binding to the glucocorticoid receptor, can directly induce DPP4 gene expression since, within the DPP4 promoter, there are two glucocorticoid-responsive elements (GREs). Interestingly, glucocorticoids highly stimulated macrophage migration through a DPP4-dependent mechanism [132]. In experimental mice, it was shown that the promoter region of DPP4 was hyperacetylated during and after dexamethasone treatment [133]. The upregulation of DPP4 by glucocorticoids might contribute to the hyperglycemic effect of these steroid hormones but also could play a role in COVID-19 physiopathology, considering that glucocorticoids have been identified as potential COVID-19 therapeutic agents because of their targeted anti-inflammatory effects [134].

#### 3.3.4. Effect of DPP4 Inhibitors on COVID-19 Patients

Considering the role of DPP4 on COVID-19 physiopathology, it can be speculated that inhibition of DPP4 might protect from SARS-CoV-2 infection or could benefit its clinical outcome. Several DPP4 inhibitors (i.e., gliptins) are approved worldwide, such as alogliptin [135], linagliptine [136], sitagliptine [137], saxagliptin [138], and vildagliptine [139], the latter with the exception of the United States. Others are approved only in Japan, South Korea, India, and/or Rusia, like anagliptin, evogliptin, gemigliptin, omarigliptin, teneligliptin, trelagliptin, and gosogliptin, whereas retagliptin remains in Phase III clinical trials [140]. Those approved by the European Medicines Agency (EMA) and the U.S. Food and Drug Administration (FDA) are summarized in Table 1 and Table 2, respectively. It is also remarkable that several generic drugs have been approved as DPP4 inhibitors by the EMA in the last year (Table 3). 

It should be pointed out that the direct effect of DPP4 inhibitors on preventing coronavirus infection has not been demonstrated to date. However, increasing evidence illustrates that DPP4 inhibitors have a beneficial effect on the clinical outcome of patients by reducing COVID-19 complications, improving recovery, and reducing mortality. In a multinational retrospective cohort study involving 56 large health care organizations, it was shown that the use of DPP-4 inhibitors was associated with a reduction in respiratory complications and a decrease in mortality, based on 2264 patients treated with DPP4 inhibitors only (i.e., alogliptin, linagliptin, saxagliptin, or sitgliptin) [141]. Likewise, a prospective randomized clinical trial with 263 COVID-19 patients showed that patients treated with sitagliptin for 2 days had better clinical outcomes and reduced lung infiltration than the control group [142]. Moreover, a prospective study with 89 COVID-19 but non-diabetic patients demonstrated that sitagliptin improved clinical outcomes, radiological scores, and inflammatory biomarkers, pointing to a potential usefulness of DPP4 inhibitors in managing non-diabetic COVID-19 patients [143]. A meta-analysis showed that the effect of gliptins was independent of age, sex, race, and location [144]. 

However, in most cases, DPP4 inhibitor users had other medications for T2DM like metformin, renin-angiotensin system inhibitors, thiazolidinediones, diuretics, or statin, making it difficult to attribute the beneficial effect solely to DPP4 inhibitors [145]. In spite of this, the clinical outcomes of COVID-19 patients using DPP4 inhibitors only was not significantly different from that using both DPP4 inhibitors and RAS inhibitors and was notably improved from COVID-19 T2DM patients without medication [145]. A recent clinical trial to evaluate the effect of the combination of linagliptin and insulin on metabolic control and prognosis in hospitalized patients with COVID-19 and hyperglycemia revealed that the combination of treatments reduced the relative risk of assisted mechanical ventilation by 74% and improved better pre and postprandial glucose levels with lower insulin requirements, and no higher risk of hypoglycemia [146]. A different study about the impact of different antidiabetic agents on individuals with diabetes and COVID-19 showed that DPP4 inhibitors were highly possible to reduce COVID-19 mortality risk in individuals with diabetes [147].

Besides, gliptins may have a role in preventing immunopathogenesis and complications of COVID-19. For instance, gliptins exert a notably anti-inflammatory effect. In experimental models of inflammation and fibrosis, gliptins suppress macrophage activation, ameliorate inflammation, reduce cytokine production, mitigate systemic inflammatory response, and reduce microvascular thrombosis [148,149,150]. In patients with T2DM and symptomatic coronary artery disease, the addition of vildagliptin to ongoing metformin showed better glycemic control, lower inflammatory markers (IL-1β and C reactive protein), higher protective markers (adiponectin and HDL-C), and improved lipid profile compared to glimepiride/metformin therapy [151]. Likewise, it was shown that sitagliptin reduced inflammation and chronic immune cell activation in HIV-infected adults [152]. In COVID-19 patients with T2DM, the use of DPP4 inhibitors reduced odds of clinical deterioration and hyperinflammatory syndrome [153]. In a recent meta-analysis addressing the potential impact of DPP4 inhibitors on COVID-19-related death, it was demonstrated that when they were administered in the inpatient setting, DPP4 inhibitors decreased the risk for COVID-19-related death by 50% [154].

Chronic inflammatory reactions may induce fibrosis by activation of myofibroblast, which produces connective tissue elements that result in substantial deposition of extracellular matrix components that progressively remodel and destroy normal tissue architecture. In fact, pulmonary fibrosis has a crucial role in COVID-19 pathology. It has been shown that DPP4 expression increases in the myofibroblasts surface when they are activated by transforming growth factor β (TGFβ) and its level correlate with myofibroblast markers and collagen deposition, suggesting a tight relationship between DPP4 and fibrosis [155]. Moreover, pharmacologic inhibition or genetic inactivation of DPP4 exerted a potent anti-fibrotic activity by notably reducing the proliferation and migration of fibroblasts, and the expression of contractile proteins [155]. Therefore, DPP4 inhibitors may be of potential use for halting progression to the hyperinflammatory and pro-fibrotic state associated with severe COVID-19 [156].

In addition, gliptins reduce macrophage infiltration to the kidney and ameliorate early renal injury [157], which indicates that DPP4 inhibitors might be a therapeutic approach to preserve renal function since renal failure is a quite common complication in COVID-19 patients. A clinical trial performed to evaluate the effects of a potent DPP4 inhibitor (gemigliptin) on kidney injury, albuminuria, and vascular inflammation among patients with diabetic kidney disease demonstrated that biomarkers of vascular calcification and kidney injury were improved significantly in the gemigliptin treatment group compared with the control group and more interesting that no serious adverse events in the gemigliptin treatment group were observed during the study [158].

In a recent Summary from Expert consensus on effectiveness and safety of DPP4 inhibitors in the treatment of patients with diabetes and COVID-19 [159], it was concluded that the use of the inhibitors may present a specific benefit in reducing mortality, particularly in in-hospital use, reducing admission to intensive care units and the need for mechanical ventilation and most importantly, the use of DPP4 inhibitors appears to be safe in patients with COVID-19. Altogether these data indicate that DPP4 inhibitors have a potential therapeutic value in the multi-organ injury caused by COVID-19.

## 4. Conclusions

Clinical data obtained when using DPP4 inhibitors showed that this protease has an influence on the risk and clinical outcome of COVID-19. Although in silico experiments that predict the compatible binding between DPP4 and S glycoprotein of SARS-CoV-2 have not been demonstrated to date in vitro or in vivo, the impact of DPP4 on the COVID-19 physiopathology goes further, due to the multiple functions developed by the protease. In this line, the proinflammatory environment developed during the course of the disease, with the hallmark cytokine storm, is also affected by DPP4 regulation over the immune system. In addition, DPP4 also endogenously controls glycemia, which appears as an striking aspect, given that type 2 diabetes has been pointed out as the most important risk factor for SARS-CoV-2 infection and one of the main comorbidities of COVID-19. To sum up, literature positions DPP4 inhibitors as candidate tools for fighting against the hyperinflammatory response typical of COVID-19. Additionally, it will be helpful for discriminating the effects of DPP4 inhibitors versus other antidiabetic drugs, to design future retrospective and epidemiologic studies in which DPP4 inhibitors could have been provided alone or having sufficient control groups for differentiating their impact on different parameters of the disease progression and prognosis. 

## Figures and Tables

**Figure 1 biomedicines-10-02026-f001:**
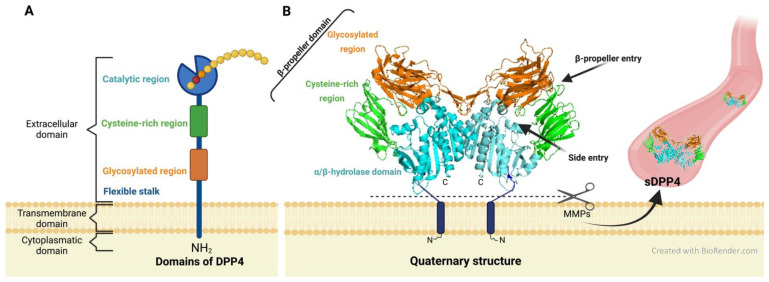
DPP4 structure. (**A**) Schematic representation of the DPP4 primary structure. The glycosylated region is shown in orange, the cysteine-rich region in green, and the catalytic region in blue. (**B**) Quaternary structure of the DPP4 homodimer. Metalloproteases (MMPs) are represented by a grey scissor, and after digestion, the soluble form of DPP4 (sDPP4) sheds from the membrane, releasing into biological fluids, such as the bloodstream (on the right).

**Figure 2 biomedicines-10-02026-f002:**
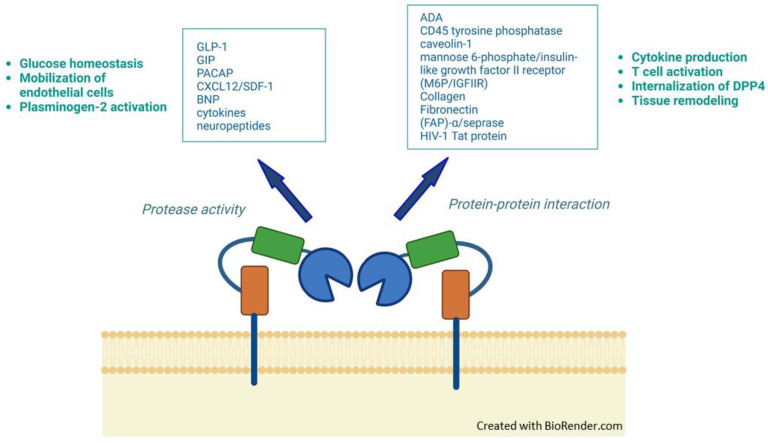
Schematic representation of DPP4 functions through its protease activity (on the **left**) or its own structure (on the **right**). Within the boxes, ligands or interacting partners of DPP4 are indicated, while the physiological processes in which they are involved are noted outside.

**Table 1 biomedicines-10-02026-t001:** List of European Medicines Agency (EMA)-approved DPP4 inhibitors.

Brand Name	Active Ingredient(s)	Marketing-Authorization Holder	EMA Product Number	EMA Approval Date ^1^
Vipidia	Alogliptin	Takeda Pharma A/S (Vallensbæk Strand, Denmark)	EMEA/H/C/002182	18/09/2013
Vipdomet	Alogliptin and metformin	Takeda Pharma A/S	EMEA/H/C/002654	18/09/2013
Incresync	Alogliptin and pioglitazone	Takeda Pharma A/S	EMEA/H/C/002178	19/09/2013
Trajenta	Linagliptin	Boehringer Ingelheim GmbH (Ingelheim and Rhein, Germany)	EMEA/H/C/002110	23/08/2011
Jentadueto	Linagliptin and metformin	Boehringer Ingelheim GmbH	EMEA/H/C/002279	19/07/2012
Glyxambi	Linagliptin and empagliflozin	Boehringer Ingelheim GmbH	EMEA/H/C/003833	11/11/2016
Onglyza	Saxagliptin	AstraZeneca AB (Stockholm, Sweden)	EMEA/H/C/001039	30/09/2009
Komboglyze	Saxagliptin and metformin	AstraZeneca AB	EMEA/H/C/002059	24/11/2011
Qtern	Saxagliptin and dapagliflozin	AstraZeneca AB	EMEA/H/C/004057	15/07/2016
Januvia	Sitagliptin	Merck Sharp & Dohme B.V. (Haarlem, Netherlands)	EMEA/H/C/000722	20/03/2007
Xelevia	Sitagliptin	Merck Sharp & Dohme B.V.	EMEA/H/C/000762	21/03/2007
Tesavel	Sitagliptin	Merck Sharp & Dohme B.V.	EMEA/H/C/000910	10/01/2008
Efficib	Sitagliptin and metformin	Merck Sharp & Dohme B.V.	EMEA/H/C/000896	15/07/2008
Janumet	Sitagliptin and metformin	Merck Sharp & Dohme B.V.	EMEA/H/C/000861	16/07/2008
Velmetia	Sitagliptin and metformin	Merck Sharp & Dohme B.V.	EMEA/H/C/000862	16/07/2008
Ristfor	Sitagliptin and metformin	Merck Sharp & Dohme B.V.	EMEA/H/C/001235	15/03/2010
Ristaben	Sitagliptin	Merck Sharp & Dohme B.V.	EMEA/H/C/001234	15/03/2010
Steglujan	Sitagliptin and ertuglifozin	Merck Sharp & Dohme B.V.	EMEA/H/C/004313	23/03/2018 (AM)
Galvus	Vildagliptin	Novartis Europharm Limited (Camberly, United Kingdom)	EMEA/H/C/000771	25/09/2007
Eucreas	Vildagliptin and metformin	Novartis Europharm Limited	EMEA/H/C/000807	14/11/2007
Xiliarx	Vildagliptin	Novartis Europharm Limited	EMEA/H/C/001051	19/11/2008
Jalra	Vildagliptin	Novartis Europharm Limited	EMEA/H/C/001048	19/11/2008
Zomarist	Vildagliptin and metformin	Novartis Europharm Limited	EMEA/H/C/001049	30/11/2008
Icandra ^2^	Vildagliptin and metformin	Novartis Europharm Limited	EMEA/H/C/001050	30/11/2008

^1^ Date (day/ month/year) of issue of marketing authorization valid throughout the European Union. ^2^ Previously vildagliptin/metformin hydrochloride Novartis. AM, additional monitoring. It means that this drug has more intense surveillance than other medicines.

**Table 2 biomedicines-10-02026-t002:** List of USA Food and Drug Administration (FDA)-approved DPP4 inhibitors.

Brand Name	Active Ingredient(s)	Company	FDA No.	Approval Date ^1^
Nesina	Alogliptin	Takeda Pharma USA	022271	25/01/2013
Kazano	Alogliptin and metformin	Takeda Pharma USA	203414	25/01/2013
Oseni	Alogliptin and pioglitazone	Takeda Pharma USA	022426	25/01/2013
Tradjenta	Linagliptin	B.-I. Pharmaceuticals, Inc. (Iowa, IA, USA)	201280	02/05/2011
Jentadueto	Linagliptin and metformin	B.-I. Pharmaceuticals, Inc.	201281	30/01/2012
Glyxambi	Linagliptin and empagliflozin	B.-I. Pharmaceuticals, Inc.	206073	30/01/2015
Jentadueto XR	Linagliptin and metformin extended release	B.-I. Pharmaceuticals, Inc.	208026	27/05/2016
Januvia	Sitagliptin	Merck & Co., Inc. (Kenilworth, NJ, USA)	021995	16/10/2006
Janumet	Sitagliptin and metformin	Merck & Co., Inc.	022044	30/03/2007
Janumet XR	Sitagliptin and metformin extended release	Merck Sharp & Dohme Corp.	202270	02/02/2012
Steglujan	Sitagliptin and ertuglifozin	Merck Sharp & Dohme Corp.	209805	19/12/2017
Onglyza	Saxagliptin	Bristol-Myers Squibb Co. (New York, NY, USA)	022350	31/07/2009
Kombiglyze XR	Saxagliptin and metformin extended release	Bristol-Myers Squibb Co.	200678	05/11/2010
Qtern	Saxagliptin and dapagliflozin	AstraZeneca Pharmaceuticals LP (Wilmington, DE, USA)	209091	27/02/2017

^1^ FDA approval date (day/month/year) is indicated. Takeda Pharms USA, Takeda Pharmaceuticals USA., Inc.; B.-I., Boehringer-Ingelheim.

**Table 3 biomedicines-10-02026-t003:** European Medicines Agency (EMA)-approved generic medicines for DPP4 inhibition.

Brand Name	Active Ingredient(s)	Generic	Company	EMA No. ^1^	Approval Date ^2^
Sitagliptin SUN	Sitagliptin fumarate	Januvia	Sun Pharmaceutical Industries Europe B.V. (Hoofddorp, Netherlands)	005741	09/12/2021
Sitagliptin/Metformin hydrochloride Mylan	Sitagliptin hydrochloride monohydrate and metformin hydrochloride	Janumet	Mylan Ireland Limited (Dublin, Irland)	005678	16/02/2022
Sitagliptin Accord	Sitagliptin	Januvia	Accord Healthcare S.L.U. (Barcelona, Spain)	005598	25/04/2022
Sitagliptin/Metformin hydrochloride Accord	Sitagliptin and metformin hydrochloride	Janumet	Accord Healthcare S.L.U.	005850	− ^3^
Vildagliptin/Metformin hydrochloride Accord	Vildagliptin and metformin hydrochloride	Eucreas	Accord Healthcare S.L.U.	005738	24/03/2022 (AM)

^1^ EMA product number should be indicated as follows: EMEA/H/C/XXXXXX. ^2^ Date of issue of marketing authorization (day/ month/year) valid throughout the European Union. ^3^ A positive opinion recommending the granting of a marketing authorization for this drug was adopted on 19/05/2022. AM, additional monitoring. It means that this drug is more intensively monitored than other medicines.

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
