# Peer review of "Role of Dipeptidyl Peptidase-4 (DPP4) on COVID-19 Physiopathology"

_biomedicines, 2022, doi:10.3390/biomedicines10082026_

Round 1
Reviewer 1 Report
The manuscript presented by Alba Sebastián-Martín et al. entitled “Role of dipeptidyl peptidase-4 (DPP4) on COVID-19 physiopathology” is well written, clear, and easy to read. The topic is of great interest considering that after two years of pandemic, we still need therapeutic options more than vaccines that could modulate our immune system properly.
Major
Please add, in the methodological procedure, which searching motors you used for the review: Pubmed, Scopus, WoS.
In the discussion section, try to open it to the other types of drugs used in contrasting covid disease, including acetaminophen, ibuprofen dexamethasone, and vitamin D isomers, and their effect on DPP4 expression. In any case, I think the manuscript is of note.
Minor:
Please, check misspellings; see lane 548 underneath table 2.
Author Response
Thank you very much for your comments and your review of our work.
According to the reviewer's suggestion, we have added a methods section indicating the searching motors used in the revision.
We agree with the reviewer that considering the effect of other drugs on DPP4 expression would improve our work. Accordingly, we have revised the effect of acetaminophen, ibuprofen dexamethasone, and vitamin D isomers, on DPP4 expression. We have found that the literature is scarce regarding all drugs but glucocorticoids. Therefore, we have added a paragraph about the effect of glucocorticoids on DPP4 expression in page 11 of the revised version of the manuscript
Reviewer 2 Report
In this review article, the authors first reviewed the structures and functions of dipeptidyl peptidase-4 (DPP4), then the potential role of DPP4 in facilitating SARS-CoV2 viral entry, and then the interactions between DPP4 and diabetes, and DPP4 and inflammation for CoVID-19 patients. Last they discussed how DPP4 inhibitors can reduce risk of COVID-19 symptoms and severe outcomes. I think the review was very well written and the topic is very interesting. I support the publication of this work in Biomedicines. I only have one suggestion: could the authors discuss the similarity between the Spike protein of SARS-CoV2 and the Spike protein of MERS-CoV? Could the authors also discuss the (potential) interaction between DPP4 and the S-protein of SARS-CoV?
Author Response
We acknowledge the reviewer for the comments on our work, for supporting the publication of this work in Biomedicines, and for the time spent in the review process. We have tried to answer the suggestion pointed by the referee. Accordingly, we have added a paragraph discussing the differences and similarities between the spike proteins of these coronaviruses, which can be found on pages 6 and 7 of the revised version of the manuscript. We have also added a comment on the possible binding of Sars-CoV to DPP4 on page 7, lines 298-300 of the revised version of the manuscript.
We hope that we have answered all the suggestions made by the reviewer and we thank he/her again for the revision.